# Psychological Mechanism in School Sports Policy Attitudes among Chinese College Students—Based on the Study of Sunshine Sports Policy

**DOI:** 10.3390/ijerph192214888

**Published:** 2022-11-12

**Authors:** Liping Liu, Shanping Chen, Xueyan Yang, Yuqing Yang, Chunyan Liu

**Affiliations:** 1Department of Physical Education, Xi’an Jiaotong University, Xi’an 710049, China; 2School of Public Policy and Administration, Xi’an Jiaotong University, Xi’an 710049, China

**Keywords:** school sports, sports policy, policy attitude, policy satisfaction

## Abstract

(1) Background: To quantitatively analyze the content structure and psychological mechanism of Chinese college students’ school sports policy attitude; (2) Methods: This paper uses the data of 2112 college students obtained from a national sample survey. We also use a theoretical model of school sports policy attitude empirically tested by the LISREL structural equation model method; (3) Results: ① Students’ cognition of the content, implementation, effect, satisfaction, and policy behavior of school sports policies are important factors affecting the implementation of school sports policy. ② Cognition generates emotions and students’ awareness of school sports policy. The higher the recognition of content, policy implementation, and policy effect, the higher their satisfaction with the school sports policy. ③ Emotion determines behavior. The greater the students’ satisfaction with the school sports policy, the more willing they are to follow the school sports policy requirements and demonstrate positive behavior in line with its provisions. ④ The measurement model of the school sports policy attitude has good reliability and validity, and the theoretical hypothesis of the structural model is supported by the data that measures and explains the college students’ attitudes toward school sports policy.

## 1. Introduction

The goal of education in China is to cultivate national builders and successors well-rounded in moral, intellectual, physical, aesthetic, and labor aspects; however, for a long time, students have emphasized academic learning and neglected physical exercise. Several national surveys on students’ physical fitness and health have established that their physical fitness level is decreasing [1]. In response to this discouraging situation, the participants of the 2006 Chinese government conference on school sports proposed a “national sunshine sport for hundreds of millions of students”. To fully implement the education policy of the Communist Party of China and the guiding ideology of “health first”, on 12 December 2006, the Chinese Ministry of Education, the State General Administration of Sports, and the Central Committee of the Communist Youth League jointly issued the “Notice on the Launch of the National Sunshine Sports for Billions of Students,” deciding to make a 2007 launch of the National Sunshine Sports for Billions of Students and widely implement the Sunshine Sports for Students (Sunshine Sports for short) [2] in schools of all levels and types. To achieve the goal of “Strengthening Teenagers’ fitness and Promoting Their Healthy Growth” the Chinese government has issued a series of school sports policies at the national level, such as the notice on Several Opinions on Strengthening School Sports [3] and opinions on Strengthening Teenagers’ Sports and Enhancing Teenagers’ Physical Fitness [4]. Sunshine Sports is an important strategic measure to strengthen teenagers’ sports and enhance teenagers’ fitness in China’s new era.

Implementing these school sports policies is critical for school sports. According to management scholars, “the effective implementation of public policies depends to a large extent on the attitudes of the subjects to whom they are applied [5]”. Guided by the Chinese government’s emphasis on “people-oriented” research, the subject of policy application has become a focal point in the field of public policy, which has increased rapidly in recent years. Research on policy attitudes of policy-applicable subjects has mainly focused on other areas of public policy and different groups of people. There are only sporadic explorations on sports policy attitudes, and the theoretical foundation is weak. To further analyze students’ attitudes toward school sports policy, previous research has constructed a school sport policy attitude-rooted theory through qualitative research methods [6], which proposed the hypothesis of the content structure and psychological mechanism of school sports policy attitude, but without empirical data support. The main purpose of this paper is to analyze quantitatively and empirically test the theoretical model using a structural equation modeling approach to provide a more sophisticated theoretical framework and measurement tools for school sport policy research and practice.

## 2. Literature Review

From the perspective of policymakers and executors, we have conducted in-depth research on the connotations, principles, instruments, processes and procedures, status and role of policies, and the factors affecting effective policy implementation. Fruitful results have promoted the development of policy science. However, relatively little research has been conducted on the subjects of policy application, especially research on the impact of the subject of policy application on policy implementation, which has not attracted enough attention in academic circles [7]. As a part of public policy, the current situation and research trends on school sports policy show the same characteristics. From the relevant research of sports policy, scholars and practitioners have focused on policy formulation and policy objectives (Sports Policy → Physical Health), ignoring certain intermediate links between the two (Sports Policy → School Implementation of Policy → Policy Attitude of Applicable Subjects → Individual Sports Behavior → Physical Health → National Development and Security), which is a casual chain from macro to micro, involving management, pedagogy, psychology, and many other disciplines.

International research on sports policy focuses on three main knowledge groups: sports politics and policy, mass fitness sports policy, and elite sports development policy [8]. China’s sports policy research has made significant progress in recent years [9], mainly involving the essence, value, policy optimization, implementation process, and execution. The research scope is incomplete, and the theoretical research lags [10]. These studies in sports management generally tend to the macro analysis of sports policy. Research in school physical education focuses on how to intervene in students’ exercise behavior through physical education, promote students’ physical health, and the effect of sports on physical health [11]. A few studies involve implementing school physical education policies and influencing factors [12]. Currently, sports psychology research mainly focuses on the persistence of sports behavior, stages of change, exercise motivation, cognitive decision-making, and the impact of sports on mental health [13]. However, influential factors do not involve school sports policy. These studies have been conducted from the perspective of macropolicy and microbehavior. They have made significant progress in a single disciplinary direction, with relatively isolated research in each discipline and a lack of cross-integrated research from a multidisciplinary perspective. The existing research still lacks research on the action mechanism from macropolicy to microindividual in sports policy implementation [14,15]. The impact and action mechanism of sports policy on students’ sports behavior remains unclear.

All public policies attempt to influence and control people’s behavior or make people’s actions consistent with the rules or goals the government sets. If the subject of policy application ignores it or opposes relevant policy requirements, it leads to the invalidity or failure of the policy. Therefore, how much influence the subject of policy application will have on policy implementation and whether it can be effectively implemented depends to a considerable extent on the attitude of the subject of policy application. Public policy research is now paying attention to the subject of policy application. Research on the policy attitude of the subject of policy application has involved fields such as childbirth [16], wages [17], benefiting farmers [18], employment [19], and education [20]. Most of the research is aimed at farmers, workers, people of childbearing age, etc., and rarely involves students’ attitudes toward public policy. College students are a special group. They think independently, do not follow blindly, and have strong self-awareness. Mature physical and mental development, growth, and development are in the golden age, with values and morality gradually maturing and stabilizing. Students’ psychological mechanisms of attitudes toward policies may differ from those of other groups.

The current situation of students’ fitness and the perspective of current school sports policy has led the state to issue the highest standard school sports policy since the country’s founding. Among them, the 2006 Sunshine Sports Policy is an important activity in comprehensively implementing the spirit of these school sports policy documents. Many schools have formulated specific measures for implementing Sunshine Sports according to their resources and characteristics. For students, Sunshine Sports is the school sports policy with which students have the most contact. After Sunshine Sports was put forward, a large number of relevant policy documents were issued by the state. Supporting policy documents on Sunshine Sports in various provinces and cities raised the importance of promoting teenagers’ fitness to an unprecedented level. However, students’ attitudes and behaviors are not very positive, and there is only temporary enthusiasm when the policy is vigorously mobilized. The improvement of students’ fitness is still not optimistic. It is evident policymakers and executors of school sports policy do not know much about the student’s policy attitudes because the implementation of school sports policy is currently ineffective. Therefore, this study focuses on college students’ attitudes towards the Sunshine Sports policy and the understanding that these specific policy documents and implementation methods have strong currency and practical value for school sports management and sports policy.

## 3. Theories and Assumptions

### 3.1. Theoretical Basis and Related Concepts

Social psychology defines “attitude as an evaluative statement about objects, people and events, either approving or disapproving. It reflects a person’s internal feelings about an object, including cognition, emotion, and behavior [21]”. Policy attitude is “the psychological tendency of satisfaction or dissatisfaction, approval or disapproval reflected by people in varying degrees when it comes to public policy [22]”. School sports policy is a specific action plan formulated for implementing education policy and sports policy, aiming at the sports problems in the field of school education. School sports policy is not only a general education policy and sports policy but also a public policy [23]. It can be seen that policy attitude is an upper concept of school sports policy attitude. According to the above definition of policy attitude, school sports policy attitude can be defined as the attitude of the subject of school sports policy application (including students, teachers, etc.) to school sports policy, including their cognition, emotion, behavioral tendency, and other psychological factors of school sports-related policies.

Classical psychology theory holds that an individual’s attitude includes three theoretical structures: cognition, emotion, and behavioral tendencies [24]. Cognition refers to the beliefs and opinions people hold about the object of the attitude, which is the basis for forming attitudes. Emotion refers to the individual’s positive or negative evaluation of the attitude object and the emotions and emotional experience caused by it. Emotion is the core of the attitude, which is closely connected with people’s behavior, and plays a regulating role. The behavioral tendency is the individual’s affirmation or denial of the attitude object. The behavioral tendency is not behavior. Instead, it is the ideological tendency before the behavior, also known as the behavioral readiness state or what reaction the individual is prepared to make to the attitude object. Once it occurs, it will have a different impact on the attitude object, affecting its direction and mode of behavior. The corresponding school sports policy attitude also includes three aspects: school sports policy cognition, school sports policy emotion, and school sports policy behavior. According to the definition of the three theoretical structures of attitude (cognition, emotion, and behavior) [25,26], the three theoretical structures of school sports policy attitude can be defined as: ① School sports policy cognition, referring to the policy application subjects’ (teachers, students, etc.) beliefs and views on the school sports policy reflected in the identification and judgment of the relevant content of the school sports policy; ② School sports policy emotion, referring to the psychological reaction of the policy-applicant to the school sports policy, expressed as satisfaction or dissatisfaction, like or dislike; ③ School sports policy behavior, referring to the relevant behavioral response of the policy-applicable subject to the school sports policy, expressed as the willingness or unwillingness to participate in activities related to the school sports policy.

In previous qualitative research, according to the content classification framework of public policy analysis [27], students’ cognition of school sports policy is divided into three categories: cognition of policy content, cognition of policy implementation, and cognition of policy effect [28]. First, according to the four indicators of policy content analysis [29], the cognition of school sports policy content includes ① necessity (whether the formulation of school sports policy is necessary), ② importance (whether the significance of school sports policy is significant), ③ rationality (whether the content of school sports policy is reasonable), and ④ feasibility (whether school sports policy provisions are feasible). Second, according to student interview data, they judge the implementation of school sports policy from five aspects: ① implementation (whether the school often organizes sports policy-related activities), ② activity form (whether the relevant activities organized by the school meet the needs of students), ③ fun of activities (whether the relevant activities organized by the school are fun and attractive), ④ time guarantee (whether students have time to participate in relevant activities), and ⑤ degree of importance (the degree to which the school attaches importance to the implementation of the policy). Third, students’ cognition of the effect of school sports policy includes four indicators, namely: ① sports opportunities (whether policy implementation provides students with more opportunities to exercise), ② physical health (whether policy implementation promotes students’ physical health), ③ exercise awareness (whether policy implementation has enhanced students’ exercise awareness), and ④ mental health (whether policy implementation has promoted students’ mental health) [13].

In related research on policy attitude, scholars have used “policy satisfaction [30,31]” to express the emotional response of the subject of policy applied to policy and rarely use the concept of policy emotion. According to convention, in our research, students’ emotional response to school sports policy is termed “school sports policy satisfaction,” which is consistent with the terminology used in the existing public policy attitude research. After students identify and judge the content, implementation, and effect of school sports policy, they will have corresponding emotional reactions. Therefore, school sports policy satisfaction includes three indicators: content satisfaction, implementation satisfaction, and effect satisfaction. According to the analysis of literature and previous research data, students’ sports policy behavior includes four indicators, namely: ① policy compliance (whether they are willing to comply with the provisions of school sports policy), ② daily exercise (whether to respond to the policy and carry out daily physical exercise), ③ activity participation (whether they are willing to participate in relevant activities organized by the school), and ④ management services (whether they are willing to undertake the management and service of policy related activities). The hierarchy of content is shown in Figure 1. 

### 3.2. Theoretical Model and Its Research Assumptions

In general, the three components of attitude are closely related. Any change in one component causes corresponding changes in the other two components, and each component is consistent with the other. According to the theory of cognitive psychology (Stimulation → Brain Processing → Response), the individual brain is an information system processing external stimuli. Individuals form judgments and evaluations of specific things in the cognitive process. These judgments and evaluations cause individual psychological emotions and emotions, form corresponding behavioral intentions based on these evaluations and judgments, and then produce corresponding behavioral responses. Therefore, attitude is an individual’s cognitive, emotional, and behavioral tendencies to specific things when dealing with external stimuli, gradually accumulating to form a stable psychological and behavioral response model. It is a product of the brain’ information processing. Based on these cognitive psychology points of view, there is a certain logical relationship between the three components of attitude. The applicable subject of policy makes policy emotion judgments based on the cognitive level, and policy emotion will impact the behavior tendency of policy. The three components of attitude are generally coordinated, and the higher the degree of coordination, the more stable the attitude. However, the three components of attitude are not always interrelated and coordinated. When the three contradict, emotional factors often play a leading role [32].

Therefore, the relationship between the five parts of school sports policy attitude can be shown in Figure 2. 

The relationship between the five main components of school sports policy attitude includes three causal relationships: Cognition → Emotion → Behavior. The hypotheses to be tested empirically are:

**H1.** 
*The higher the students’ satisfaction with the school sports policy, the more willing they are to follow the policy requirements and show positive behavior in line with its provisions.*


**H2.** 
*The higher students’ recognition of school sports policy content, the higher their satisfaction with it.*


**H3.** 
*The higher the students’ recognition of the school sports policy implementation process, the higher their satisfaction with it.*


**H4.** 
*The higher students’ recognition of the implementation effect of the school sports policy, the higher their satisfaction with it.*


## 4. Research Methods

### 4.1. Research Objects

Undergraduates were the research object. The stratified random sampling method was used to select the survey objects nationwide. First, 20 provinces and urban areas were randomly sampled from 31 provinces, municipalities directly under the central government, and autonomous regions in the Chinese Mainland. A representative school was selected in each region for investigation. Secondly, according to the needs of the research design for the total sample size, 120 college students were selected from each of the selected 20 colleges and universities. To reasonably distribute the samples in grade and gender, the collaborators of the cooperative colleges and universities were required to conduct stratified random student sampling. As a result, 30 students (15 men and 15 women) were randomly selected from each of the freshman, sophomore, junior, and senior grades. During the survey, 2400 questionnaires were distributed, and 2276 questionnaires were recovered, with a recovery rate of 94.8%. After the questionnaire review and logical verification, 164 invalid questionnaires were excluded. The final valid data were 2112, including 543 in grade one, 562 in grade two, 520 in grade three, and 487 in grade four; 1037 men and 1075 women. See the flow chart in Figure 3 for details.

### 4.2. Measurement Tools

According to Figure 1, “content framework of school sports policy attitude,” a second-order measurement model is designed for school sports policy attitude. The second-order factor is school sports policy attitude, and the first-order factor is five main components. The questionnaire is designed according to the specific index content of the five components, and each index is designed with a topic, which is a statement with the nature of identification. For example, the importance index of policy content cognition is “the significance of sunshine sports policy is very significant.” The implementation index of policy implementation cognition is “schools often organize sunshine sports-related activities.” The health index of policy effect cognition is “the implementation of sunshine sports has promoted my health.” The effect satisfaction index of policy satisfaction is “I am satisfied with the implementation result of Sunshine Sports policy.” The policy compliance index of policy behavior “I am willing to comply with the regulations and requirements of sunshine sports policy.” To avoid the consistency tendency of subjects’ answers caused by the concentration of questions of the same factor, the questions are disordered and randomly arranged in the practical questionnaire. At the same time, to test whether the students answer the questions carefully and facilitate logical verification when cleaning the data, some questions are set as the content of reverse answers. Five-level Likert measured all questions, and the five alternative answers were “agree,” “agree more,” “uncertain,” “disagree more,” and “disagree.” The scores are 5, 4, 3, 2, and 1, respectively, as the topic scores. Some topics are reverse scores, while the five-dimension and total scores are the average measurement topic scores. The higher the score, the more positive the students’ attitude towards school sports policy.

To ensure content validity, the measurement topic was revised and reviewed multiple times by six experts with work experience and research experience in school sports, public administration, and psychology. They confirm it can reflect the operational definition of the concept of school sports policy. After the scale was designed, three colleges and universities were selected for a trial survey with 600 students randomly sampled by grade and gender. Two weeks later, 44 students were measured a second time. These data were used to test the reliability and validity of the scale. The internal consistency reliability coefficient (Cronbach a) of the five subdimensions of the scale was 0.744–0.845, and the retest correlation coefficient was 0.541–0.665.

### 4.3. Data Analysis Methods

Structural equation modeling (SEM, LISREL 8.53 software) was used to analyze the survey data of 2112 samples to test the psychological mechanism hypothesis of the relationship between the five components of attitude.

## 5. Results and Analysis

According to assumptions advanced by the school sports policy attitude theory, the statistical model of LISREL structural equation is constructed. The maximum likelihood method is used to analyze the data in LISREL software. The basic model of the structural equation of LISREL software analysis results is shown in Figure 4. The structural equation model takes the five components of school sports policy attitude as latent variables (oval legend in Figure 4). Four paths representing causality are set between the five latent variables (arrow in Figure 4). The explicit variables of the structural equation model have 20 direct measurement indicators, including four indicators of policy content cognition, five indicators of policy implementation cognition, four indicators of policy effect cognition, three indicators of policy satisfaction, and four indicators of policy behavior (rectangular legend in Figure 4). The statistical results include the fitting index, measurement model, and structural model, which are analyzed below.

### 5.1. Fitting Index

Table 1 shows the goodness-of-fit analysis results of the structural equation model of school sports policy attitude theory. The *RMSEA* (root mean square of approximate error) of the theoretical structural model of school sports policy attitude is 0.095, which reaches the better fitting level (*RMSEA* less than 0.1) recognized by Steiger, the builder of the goodness-of-fit index, indicating that the gap between the structural equation model and the saturated model is relatively small.

*CFI* and *NFI* are relative fitting degrees, and the theoretical structure model of school sports policy attitude is 0.96, higher than Bentler’s suggested level of 0.90. The model’s *NFI* is 0.96, which is higher than the level of 0.90 suggested by Bentler and Bonett [33], indicating that the fitting degree of the theoretical model to the data is 96% better than the void model. This suggests that the theoretical model of school sports policy attitude is greatly improved compared to the void model and fits the data very well.

*GFI* and *AGFI* are indicators of the model’s absolute suitability. The greater the difference between these two indicators, the more insignificant paths are included in the structural equation model. In Table 1, the difference between the *GFI* and *AGFI* of the structural equation model of school sports policy attitude is 0.04, indicating that the path coefficient in the structural equation model is highly significant.

*PGFI* and *PNFI* are indexes derived from *GFI* and *NFI* to save the fitting degree, which mainly reflects the trade-off relationship between model fitness and degree of freedom. The application of this index does not advocate complex models (models with less degrees of freedom) [34]. The PGFI and *PNFI* of the theoretical model of school sports policy attitude are higher than the standard of 0.50 identified by James, the index builder, indicating that the structural model of school sports policy attitude conforms to the principle of simplicity.

### 5.2. Measurement Model

Table 2 shows the complete standard solutions (lambda-y and lambda-X) of the measurement model in the theoretical structural equation of school sports policy attitude. The table contents include the number of indicators (measurement topics) contained in the latent variables (five factors of school sports policy attitude), the factor load of the relationship between the latent variables and indicators, and the correlation coefficient matrix between the latent variables. The factor load of the measurement model meets the standard of 0.5. Although the load value of individual factors (the third indicator agreed by the policy implementation: time guarantee) is low, it is also close to the acceptable standard of 0.4. The high factor load of the measurement model reflects that the questions of the school sports policy attitude scale have good convergent validity.

The second half of Table 2 is the correlation matrix of the scores of five factors of school sports policy attitude. The data in the correlation matrix represents the correlation between potential variables. The absolute value of the correlation coefficient in Table 2 is between 0.44 and 0.74, in a reasonable range, indicating that the measurement questions of the five factors of school sports policy attitude have good discriminant validity.

### 5.3. Structural Model

According to the theoretical hypothesis put forward by the psychological mechanism of school sports policy attitude, LISREL software is used to set up four paths between latent variables for the structural equation model of school sports policy attitude. Table 3 gives the standardized estimated values of the path coefficients of the structural model of LISREL software’s attitude toward school sports policy. The results show that the *t* value of the four standardized path coefficients in the structural equation model reaches 0.001. The statistical results show that the theoretical hypothesis of the psychological mechanism expressed by the theoretical model of school sports policy attitude is tenable.

For the result variable of the theoretical model of school sports policy attitude–school sports policy behavior, the path coefficient of the direct antecedent variable school sports policy satisfaction is 0.86. This is significant (*p* < 0.001), with the variance explanation of policy satisfaction to school sports policy behavior reaching 0.61. The statistical results show that there is a significant positive correlation between school sports policy satisfaction and school sports policy behavior, which supports hypothesis H1, namely:

**H1.** *The higher students’ satisfaction with the school sports policy, the more willing they are to follow the policy requirements and show positive behavior in line with its provisions*.

For the core variable of the theoretical model of school sports policy attitude–school sports policy satisfaction, the influence of the three cognitive variables proposed by the theoretical model is very significant (*p* < 0.001). The causal relationship between the path coefficient of school sports policy satisfaction and the theoretical hypothesis is completely consistent with the cognition of school sports policy content (0.57), the cognition of school sports policy implementation (0.19), and the cognition of school sports policy effect (0.31). These three antecedent variables have a significant positive impact on school sports policy satisfaction. The explanation degree of variance change of the three antecedent variables on policy satisfaction is 0.64, among which the cognition of school sports policy content is the most influential factor among the three antecedent variables. Its effect on policy satisfaction is three times that of school sports policy implementation cognition and 1.84 times that of school sports policy effect cognition. This result supports another three theoretical hypotheses of the psychological mechanism of school sports policy attitude:

**H2.** 
*The higher students’ recognition of school sports policy content, the higher their satisfaction with it.*


**H3.** 
*The higher students’ recognition of the school sports policy implementation process, the higher their satisfaction with it.*


**H4.** 
*The higher students’ recognition of the implementation effect of the school sports policy, the higher their satisfaction with it.*


## 6. Discussion

### 6.1. Evaluation of the Theoretical Model

The theoretical model of school sports policy attitude includes the theoretical content framework and psychological mechanism. The evaluation of these two parts of the theoretical model is reflected in the relevant tests of the measurement model and structural model.

The evaluation of the theoretical model includes data fit, model simplicity, local parameter significance, and the interpretation ability of the theoretical model. A good theoretical model should be concise and consistent with the data. From the structural equation model of school sports policy attitude and the data fitting index, the structural equation model is supported by the survey data. The RMSEA in Table 1 is 0.095, reaching a good fitting level, indicating that the gap between the structural equation model and the saturation model is relatively small; both NFI and CFI are 0.96, exceeding the very good fitting standard of 0.95, reflecting that the structural equation model of school sports policy attitude has much improvement over the void model, and fits the data very well. From the perspective of the simplicity of the theoretical model, the PGFI and PNFI (saving fit) of the structural equation model of school sports policy attitude are higher than the level of 0.50, indicating that the theoretical model is very concise. Regarding the significance of local parameters, the path coefficients between the four latent variables set by the theoretical model are all significant (*p* < 0.001). From the perspective of the explanatory ability of the structural equation to the core variables, the antecedent variables of the theory can explain and predict the outcome variables well. The explanation of students’ school sports policy satisfaction to the variance variation of school sports policy behavior is 0.61, and the explanation degree of the three policy cognitive variables to the variance variation of policy satisfaction is 0.64, indicating that the predictive ability of the theoretical model of school sports policy attitude is very strong. Overall, the theoretical model can effectively predict and explain college students’ attitudes toward school sports policy and has good reliability and validity.

### 6.2. The Impact of Students’ Policy Attitudes on Policy Implementation

Public policy uses measures to influence and guide the behavior of the subject of policy application, so that their behavior is consistent with the government’s goal. The expected desired behavior will only occur if a policy is clearly communicated to the students [35]. The main goal of school sports policy is to guide students to participate in physical exercise actively and promote students’ physical and mental health. For example, Guo Ban Fa No. 27 mentioned “taking daily exercise, healthy growth and lifelong benefit as the goal [36]”. From the perspective of these objectives of school sports policy, the most direct and explicit impact on the implementation of school sports policy is mainly the policy behavior of students, such as whether students comply with the provisions and requirements of the “Sunshine Sports” policy and exercise for 1 h daily, whether students do their best in the physical fitness test, and so on. These concerns are directly related to the goal of the school sports policy, which is also of concern when the school sports policy is implemented. Relatively speaking, students’ cognition and emotion about the policy are implicit and indirect to the policy objectives. Consequently, they are often ignored in the policy implementation process. According to the theory of attitude in psychology, cognition produces emotion, and emotion determines behavior. The higher students’ evaluation of the school sports policy, the higher their satisfaction with it. The higher students’ satisfaction with the school sports policy, the more willing they are to follow its requirements and show positive behavior in line with its provisions. From this empirical data, cognitive and emotional factors have played a key role in determining students’ policy behavior.

In recent years, a series of strong school sports policies in China have had a significant impact: “the downward trend of teenagers’ fitness has been suppressed, but there are still some areas that need to be further improved.” The 2015 China youth sports development report mentioned that “the physical health of middle school students has stopped ‘falling’ and gradually recovered, but the endurance quality of college students continues to decline.” E. Bardach’s policy implementation game model believes that if the policy target group is unwilling to adhere to the policy, policy implementation will be ineffective [37]. School policies will only be effective if students clearly understand them [38]. Students are the ultimate target group of school sports policy. If students’ attitude towards school sports policy is negative and they do not actively respond to the policy’s call, the effect of policy implementation will be greatly reduced. Due to strong university student autonomy, students’ attitudes will have a greater impact on the effect of policy implementation. The empirical study found that: “students’ attitude and cognitive level towards the standard test are the subjective influencing factors of the standard test results [39]” and “the continued decline of College Students’ endurance quality is largely the result of universities’ neglect of students’ policy attitude in the process of implementing school sports policies to promote students’ physical health [40]”.

### 6.3. Implications for School Sports Work

Given the current situation, school sports policy shows the phenomenon of “barber’s pick: one head hot.” On the one hand, because of the current situation of Chinese students’ fitness, the specifications of school sports policy issued by the Chinese government in recent years have been continuously improved, which has raised the importance of promoting teenagers’ fitness to an unprecedented level. On the other hand, students’ attitude is not very positive or just a fleeting enthusiasm when the policy is vigorously mobilized. Improvement in students’ fitness is still not optimistic. This phenomenon shows that, in implementing school sports policies, we only pay attention to the explicit goals of students’ fitness and exercise behavior while ignoring some implicit but key influencing factors of students’ cognition and emotion. Research has also identified policy implementation as the single most important issue related to school policy [41]. To achieve the goals stipulated in the school sports policy, we can use mandatory measures to force students to comply with the provisions and requirements of the school sports policy, but only external mandatory behavior intervention, without cognitive intervention, will create students’ internal emotional conflict and bring students negative cognition and negative emotional experience. Such an intervention effect is not sustainable. Therefore, school sports policy should pay attention to students’ policy behavior and also pay attention to students’ policy cognition and policy emotion. In the policy implementation process, we should ask students to do something (behavioral intervention) and tell students why they should do so (cognitive intervention) so that students can integrate knowledge and practice.

American scholars McCall and Weber believe that policy analysis mainly includes the study of policy content and process and the evaluation of policy implementation results, which are the three aspects of students’ cognition of school sports policy. From the theoretical model test results, the content, implementation, and effect of school sports policy significantly impact students’ satisfaction with school sports policy. Whether students follow the behavior intention of school sports policy is closely related to these three aspects. The cognitive intervention of students’ attitudes towards school sports policy should include these three aspects. Among these three cognitive variables, students’ cognition of sports policy content has the greatest impact and can be used as the key intervention content of students’ sports policy attitude. Students’ positive and negative cognition of school sports policy is related to the policy’s rationality. It is also related to educators’ communication and interpretation and students’ correct understanding. From existing research and the results of this survey, students hold a relatively positive attitude towards school sports policy. Students’ cognitive score of school sports policy content is the highest among the three cognitive factors (4.071 ± 0.790). The cognitive score of school sports policy implementation is relatively low (3.438 ± 0.872), indicating that specific measures and methods in the school sports policy implementation process must be improved.

Student perceptions indicate policy implementation and provide a different type of information than that described in official policy documents [41]. Students’ attitudes towards school sports policy can be used as feedback information for policy implementation. Students’ evaluation of the effect of policy implementation should be included in policy evaluation. Concerning this information, school sports policy and implementation methods conducive to students’ acceptance can be formulated to improve the effect and satisfaction of policy implementation.

## 7. Conclusions and Policy Implications

### 7.1. Conclusions

(1) Students’ attitude towards school sports policy includes five components, which are important factors affecting the implementation of school sports policy.

(2) Cognition produces emotion. The higher the students’ recognition of the content, implementation, and effect of school sports policy, the higher their satisfaction with it.

(3) Emotion determines behavior. The higher students’ satisfaction with the school sports policy, the more willing they are to follow the requirements of the school sports policy and show positive behavior in line with the provisions of the school sports policy.

(4) The measurement model of school sports policy attitude has good reliability and validity, and the theoretical assumptions of the structural model are supported by data that validly measure and explain college students’ attitudes towards school sports policy.

This paper proposes five theoretical structures of college students’ policy attitudes toward school sports policy: content cognition, implementation cognition, effect cognition, satisfaction cognition, and policy implementation. These structures enrich policy research theory, deepen understanding of college students’ school sports policy attitudes, and provide theoretical support for the smooth implementation of school sports policy. Meanwhile, they empirically confirm the causal relationship of school sports policy attitudes. The empirical evidence also confirms the causal relationship of “cognition → emotion → behavior,” which is conducive to guiding school sports work, formulating intervention channels for school sports policy attitudes, improving students’ satisfaction with school sports policies and positive behaviors in line with school sports policies, and effectively promoting college students’ physical health.

### 7.2. Revelations

The current study of policy attitudes toward school sports policy presents a model of the content structure and psychological mechanisms of policy attitudes that provide a reference for other public policy analyses. The authors argue that for different populations and different public policies, despite the existence of different policy attitudes and the specific effects of policy attitudes, the core theoretical structure of policy attitudes expressed in the theoretical model finds its basis in classical psychological and public administration theories. This model should have some universality and apply to varied populations only in the field of school sports policy. The theoretical framework can also be extended to other areas of public policy research. However, in different regions and countries, due to different policy contents, implementation processes, implementation efforts, and other factors translate into differing extensions of the theory involving measurement aspects and specific topics and observations. Adjustments are merited according to the actual situations.

### 7.3. Deficiencies

This empirical study uses a cross-sectional study. The analysis of data from one survey can only confirm the existence of significant correlations among these elements. Whether the intervention approach proposed based on the study results can effectively change students’ attitudes toward school sports policy requires experimental research testing with multistage longitudinal data. The Sunshine Sports policy introduced by the Chinese government is oriented toward students of all academic levels. The subjects of this study were college students. Although the sample size is sufficiently large, the subject of policy application is still relatively single, and the college student sample does not yet represent all students. The external validity of the theoretical hypothesis has limitations, so it is recommended to further test and revise it for other characteristics of student groups in the future, including secondary and elementary school students.

## Figures and Tables

**Figure 1 ijerph-19-14888-f001:**
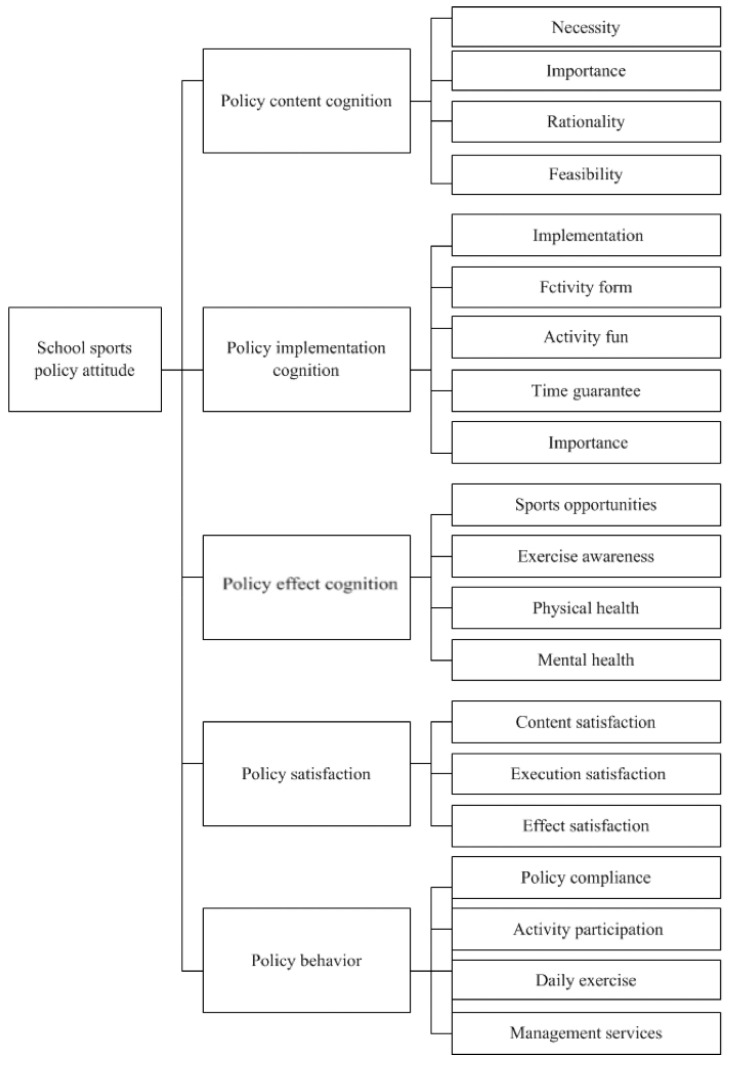
Content framework of school sports policy attitude.

**Figure 2 ijerph-19-14888-f002:**
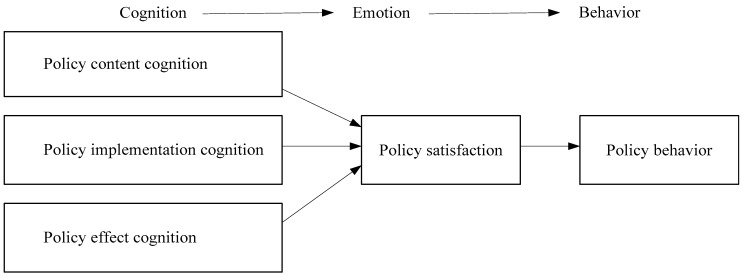
Schematic diagram of psychological mechanism of policy attitude.

**Figure 3 ijerph-19-14888-f003:**
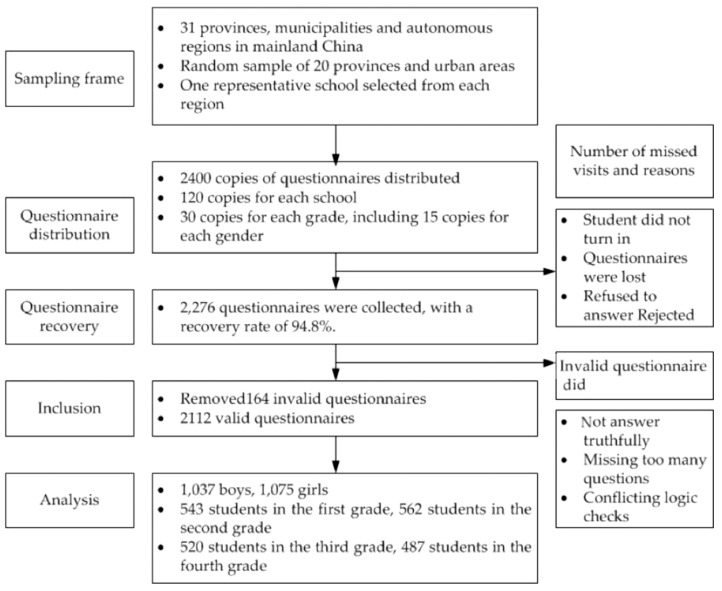
Flow chart.

**Figure 4 ijerph-19-14888-f004:**
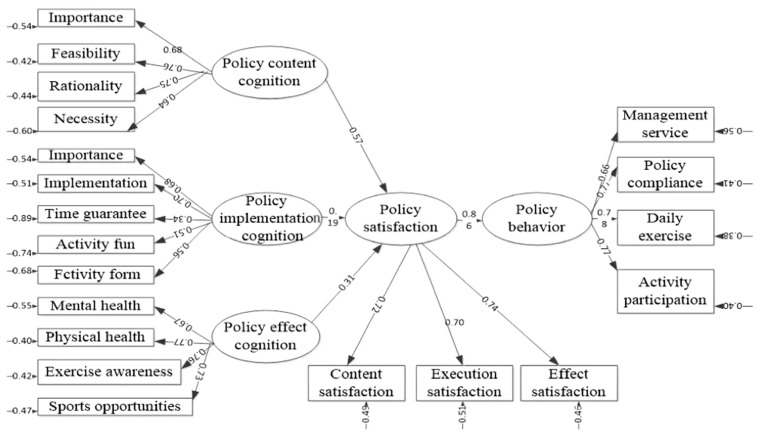
Structural equation model of school sports policy attitude theory.

**Table 1 ijerph-19-14888-t001:** Goodness of fit statistics of structural models.

*X^2^*	*df*	*RMSEA*	*CFI*	*NFI*	*GFI*	*AGFI*	*PGFI*	*PNFI*
2801.52	163	0.095	0.96	0.96	0.87	0.83	0.67	0.82

**Table 2 ijerph-19-14888-t002:** Factor load value and correlation matrix of school sports policy attitude measurement items.

Latent Variable	Number of Items	1	2	3	4	5	Policy Content	Policy Implementation	Policy Effect	Policy Satisfaction	Policy Behavior
Policy content cognition	4	0.68	0.76	0.75	0.64		1.00				
Policy implementation cognition	5	0.68	0.70	0.34	0.51	0.56	0.50	1.00			
Policy effect cognition	4	0.67	0.77	0.76	0.73		0.73	0.54	1.00		
Policy satisfaction	3	0.72	0.70	0.74			0.72	0.58	0.67	1.00	
Policy behavior	4	0.66	0.77	0.78	0.77		0.73	0.44	0.74	0.60	1.00

**Table 3 ijerph-19-14888-t003:** Standardized path coefficient of school sports policy attitude structure model (*n* = 2112).

Model Path	Standardized Path Coefficient	T Value	*p* Value	Corresponding Assumption
1. Policy satisfaction → Policy behavior	0.86	26.76	<0.001	H1+
2. Policy content → Policy satisfaction	0.57	9.94	<0.001	H2+
3. Policy implementation → Policy satisfaction	0.19	7.42	<0.001	H3+
4. Policy effect → Policy satisfaction	0.31	5.30	<0.001	H4+

## Data Availability

The raw data supporting the conclusions of this article will be made available by the authors without undue reservation.

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
