# Peer review of "Psychological Mechanism in School Sports Policy Attitudes among Chinese College Students—Based on the Study of Sunshine Sports Policy"

_ijerph, 2022, doi:10.3390/ijerph192214888_

Round 1

Reviewer 1 Report

Dear authors, you have made an important effort to evaluate the impact of a public policy, I only suggest some minor annotations to help in the understanding of the manuscript:

In the introduction, it is suggested to specify the country's educational context, because it is assumed that readers know the country's dynamics, so contextualizing it to be better understood by readers from a country other than China would be appropriate. 

In the methodology, a flow chart may be important to facilitate the understanding of the process, when, how and how many participants, it is suggested to follow the STROBE protocol, to improve transparency and quality in the results, which I insist you have tried to be very clear. 

Revise the bold type in the paragraph that begins on line 279 and try to adjust the tables to the journal's standards. 

Include the limitations of the study, I did not find them in your discussion. 

I suggest contrasting your results with those found in the rest of the world because you discuss with a majority of Asian authors, which biases the problem towards a local discussion. 

Finally, 

How much are the results found generalizable to the rest of the world? 

Reviewer 2 Report

This manuscript quantitatively analyzed the content structure and psychological mechanisms of school sports policy attitudes of Chinese college students using a structural equation model method. The findings are valuable for the study and formulation of school sports policies. However, there are some problems needing to be solved. So I recommend this manuscript to be published after some revisions.

1. There are some detail errors in the paper, such as line 260 where the second agree more should be disagree more. Please check the content of the full paper.

2. There are problems with the regularity of the figures in the article, for example, the text in Figure 2 is oriented differently. Please strengthen the normality of the paper.

3. The conclusion does not reflect the value of the findings very well. So I suggest that the author make some minor revisions to the conclusion, especially to highlight the innovation of the article and the practicality of findings.
